# Functional Recovery after the Application of Amniotic Tissues and Methylene Blue during Radical Prostatectomy—A Pilot Study

**DOI:** 10.3390/biomedicines11082260

**Published:** 2023-08-13

**Authors:** Dimitri Barski, Igor Tsaur, Mihaly Boros, Jan Brune, Thomas Otto

**Affiliations:** 1Department of Urology, Rheinland Klinikum Neuss, 41464 Neuss, Germany; thomas.otto@rheinlandklinikum.de; 2Department for Urology and Pediatric Urology, University Medical Center of Johannes Gutenberg, 55131 Mainz, Germany; 3Institute of Surgical Research, University of Szeged, 6720 Szeged, Hungary; 4DIZG—Deutsches Institut für Zell-und Gewebeersatz gGmbH, 12555 Berlin, Germany; 5Medical School, University of Duisburg-Essen, 45147 Essen, Germany

**Keywords:** prostatectomy, incontinence, impotence, tissue engineering, amnion, MB

## Abstract

Amniotic tissues and methylene blue (MB) provide the ability for neuroregeneration, and MB enables intraoperative neurostaining. We first combined the techniques to explore a neuroprotective effect on early functional outcomes in a retrospective proof-of-concept trial of 14 patients undergoing radical prostatectomy (RP). The patients were followed up at a median of 13 months, and the continence and potency rates were reported. Early recovery of continence was found after three months. No effect on potency was detected. The findings indicate the feasibility of this tissue-engineering strategy, and justify prospective comparative studies.

## 1. Introduction

Nerve-sparing surgery has attracted much attention in recent years, especially in general surgical (e.g., rectal surgery), gynecologic (e.g., hysterectomy), and urologic operations. The functional outcome is an important quality-of-life parameter after radical prostatectomy (RP) for prostate cancer. There is an unmet need to improve continence and potency after pelvic surgeries such as radical prostatectomy. The postoperative continence rate, defined as no pads used, varies between 4% and 31% (mean 16%) [1]. Better early continence results are presented in patients who have undergone the robotic series of procedures. Some patients recover from incontinence after rehabilitation, but 10–20% suffer from persistent incontinence, and 20–70% from erectile dysfunction [2,3]. A better understanding of anatomy and an improvement in surgical techniques and devices have evolved over recent years. For example, a surgical robotic system delivers 3D high-definition magnified views of the surgery, and the tiny instruments increase the surgeon’s range of motion, for precise and gentle surgery. However, laparoscopic and robot-assisted RP with nerve-sparing strategies has failed to show significant benefits in oncological and functional outcomes, compared to open RP [4]. The available randomized studies and meta-analyses show a benefit for early continence and potency, but there is a lack of evidence supporting any clear benefit of the robotic procedure (RARP). A current meta-analysis found no significant difference between the robotic and laparoscopic approaches regarding continence (odds ratio (OR) 1.95, 95% confidence interval (CI) 0.67–5.62) after 12 months. However, at 3 and 6 months, there were significant differences in favor of RARP. Patients undergoing RARP consistently show better potency postoperatively than those undergoing the laparoscopic approach [5]. Surgical intervention and radiotherapy are the main reasons for scarring and nerve damage [6]. The traction on the tissue during the prostatectomy procedure causes tears in the area of the dorsal prostatic capsule and the nerve plexus. In addition, the inflammatory response triggers edema, acidosis, and apoptosis, which can potentially lead to nerve damage [7,8,9]. This causes a delay in reaching continence and potency. So far, there is no ideal strategy to prevent scarring and adhesion formation in the nervous system. The clinical benefit of growth and anti-inflammatory factors could be crucial to regenerating the neurovascular bundle (NVB) [9].

Various natural materials, such as veins, laminin, and collagen, and synthetic materials have been used in combination with stem cells to reconstruct peripheral nerves [10,11]. There are difficulties, such as adverse immune reactions, and the possible use of immunosuppressants. From anatomic studies, we know that it is important to spare the nerve plexus, which is located individually, with numerous fibers, and is challenging to visualize. Studies have explored neuropraxia during radical prostatectomy, but few use tissue engineering to protect the pelvic nerve plexus [12,13,14]. 

Amniotic membranes (AMs) could serve as a potential valuable scaffold for tissue-engineering strategies in pelvic surgeries. Our preliminary experimental and animal studies proved the regeneration potential of AMs in reconstructive urology [15,16,17]. The amniotic membrane is 0.02–0.05 mm thick; the basal membrane is robust, avascular, and contains no nerves or muscle cells. The extracellular matrix comprises collagen and elastin, allowing for a high AM tensile strength. Mesenchymal stem cells secrete numerous growth factors. The extracellular matrix of the amnion can store and secrete active molecules produced by the epithelial and mesenchymal cells. Several hundred growth factors, cytokines, chemokines, protease inhibitors, and other bioactive molecules that modulate tissue healing have been identified in the AM [9]. These include growth factors such as the epidermal growth factor (EGF), hepatocyte growth factor (HGF), and keratinocyte growth factor (KGF), which stimulate epithelial cell growth. The AM serves as an anti-microbial barrier, and is anti-inflammatory via the down-regulation of pro-inflammatory factors. The inflammatory reaction is the decisive factor in tissue regeneration. The AM contains regulating agents that coordinate the acute and chronic phases of the inflammatory response. The AM is immune-privileged fetal tissue, which is regularly used for allograft transplantation. The amnion is obtained sterilely during cesarean section, and is therefore available indefinitely. Preservation takes place in two forms, dried or frozen; this does not seem to have any clinical influence. In our work, the AM has been used as a bioactive scaffold, experimentally in rats and clinically, to create a suitable environment for regenerating defects in the bladder wall. We aimed to investigate the possibility of bladder and colon grafting, and the immunogenicity of processed AM xenografts. Histological analyses were performed to investigate the degradation of the AM, and graft rejection and the ingrowths in the surrounding tissue 7, 21, and 42 days after implantation. The trial series was a success, as tissue regeneration was initiated, and the graft was not overgrown by scar tissue [15,16,17]. The AM experimentally showed neuroprotective and anti-carcinogenic properties, which make it a potentially interesting scaffold for early functional recovery after radical prostatectomy [18,19]. 

Additionally, we developed a strategy to better visualize the nerve plexus. Within neurophysiology and anatomy, the supravital staining of nerve structures with methylene blue (MB) is a well-known procedure [20]. Nerve-fiber staining with MB is an old method, first described by Paul Ehrlich in 1885 [21]. Coers and Woolf, in 1959, elaborated on the original technique, and recommended using methylene blue for intravital nerve staining. According to these researchers, the underlying principle is that methylene blue is an antioxidant, preferably absorbed by nerve tissue, and improves the visualization of the nerve plexus [20,22]. We are the first research group to apply the combination of amniotic tissues and MB clinically during radical prostatectomy. The procedure could pave the way for other pelvic nerve-sparing surgeries. The primary aim of this proof-of-concept study was to explore the neuroprotective effect of the amnion and MB on early functional outcomes after open radical prostatectomy.

## 2. Materials and Methods

Fourteen patients with low- or intermediate-risk prostate cancer were selected from Nov 2021 to Feb 2023 for open radical prostatectomy (RP). The selection criteria were preoperatively continent patients (no pads used), between 40 and 75 years old, with clinically localized cancer (cT1–T2), who had given written consent. Patients with metastasis or signs of extracapsular disease in the preoperative imaging, a previous history of pelvic radiotherapy, or prostate cancer treatment were excluded. Other exclusion criteria were preoperative erectile dysfunction (ED, not enough for sexual intercourse in less than 50%) and non-compliance with the follow up. One patient with preoperative high-risk cancer was included. Two planned cases were excluded due to unexpected extended disease. Five patients with intraoperative venous, oozing bleeding were excluded, due to the risk of amnion displacement. Open radical prostatectomy was performed by one very experienced surgeon (>1500 open RPs). Routinely, the operation time, blood loss, and transfusion rate were determined for each patient. In low- and intermediate-risk prostate cancer, intraoperative localized prostate cancer, and preoperative continence (no pads) and potency (enough for sexual intercourse ≥ 50%), unilateral or bilateral nerve sparing was performed by leaving the seminal vesicle tips and vascular nerve bundles (NVBs) in place. For improved visualization, and the regeneration of the nerve plexus in the region of Denonvilliers’ fascia, the entire region was processed with MB (5 mg/mL, Provepharm, France, 2:8 dilution with 0.9% NaCl), using the technique previously described in [20] (Figure 1). Based on the MB visualization, an individual cutting of the prostate, with an optimal sparing of the neuro-vascular plexus, was performed.

The human amniotic membrane (AM, 3 cm × 3 cm, DIZG gGmbH, Germany) arrived dried and sterile-packed. First, the amnion was cut into three longitudinal pieces (3 cm × 1 cm). Two portions were placed around the vascular nerve bundles, and the third amniotic portion was placed at the dorsal part of the vesico-urethral anastomosis (Figure 2). The degree of nerve-sparing was determined based on preoperative histological factors, the number of positive cores, and the intraoperative findings.

The study’s primary goal was to explore the effect of tissue engineering with AM and MB on the early functional outcomes in a proof-of-concept study.

The following criteria were recorded for each patient, as part of the existing routine:
The assessment of urine leakage according to the number of used pads.The assessment of potency according to the ability to maintain an erection sufficient for sexual intercourse. Treatment of erectile dysfunction with, for example, PDE-5 inhibitors or SKAT was recorded.The time of catheter removal, using a cystogram on the 5th–7th postoperative day, and testing of the anastomosis for tightness (instillation of 200 mL iodized contrast medium). If there was evidence of contrast medium leakage, the catheter was left in place. The catheter removal was reported as *n* = x days postoperatively.The PSA course and additional oncological treatment (radiotherapy, hormonotherapy).The assessment of complications, according to the Clavien–Dindo classification.

## 3. Results

A total of 14 carefully selected patients were included and received the above treatment; 79% were classified as “intermediate risk“, and all were continent and potent preoperatively (Table 1). 

The intra- and postoperative data are summarized in Table 2. The median follow up was 13 months (range: 3–18 months). The mean operative time was 71 min (56–85 min). No major complications occurred; 79% had bilateral nerve sparing. No signs of allergic reaction or graft rejection were found.

The overall potency outcomes showed loss of interest or complete erectile dysfunction in seven patients (50%). All patients were included in this analysis, regardless of the degree of neuroprotection. The potency recovery rates at 1, 6, and 12 months were similar. 

The overall continence rate during the follow up was 93% wearing no pad, or one safety pad. Only one patient suffered from early severe incontinence and voiding dysfunction; they were still under evaluation, probably for sphincter suture and VUA stenosis. An early continence return was reported after 3 months (Figure 3). We also found improved VUA healing in cases of cystoscopy control. There were no signs of scarring but the beginning of epithelialization with urothelial mucosa after several weeks. Overall, a good postoperative quality of life was reported by the patients. One patient with locally advanced prostate cancer and multiple positive surgical margins showed no local recurrence after 6 months.

## 4. Discussion

There is a high prevalence of incontinence and erectile dysfunction after radical prostatectomy (RP) [2,3]. Tissue engineering develops as a new promising field of science to achieve better postoperative outcomes. The key role of tissue engineering is to find a scaffold to enhance tissue regeneration via the secretion of growth factors. Strategies to prevent fibrotic nerve tissue reactions, and to improve tissue healing in reconstructive urology are lacking. The AM can be used as an immunomodulatory material for intraoperative tissue engineering. This ability of the AM is mediated by a variety of cytokines, many of which are involved in tissue regeneration and the control of the inflammatory response. There are several key factors that make the AM a potential biomaterial for tissue engineering. These include immunotolerance, regenerative, anti-angiogenic, anti-fibrotic, and anti-bacterial properties. Today, AM allografts are considered a standard therapy for ocular surface reconstruction, as demonstrated in several randomized and controlled trials [23]. The efficacy of the AM has been demonstrated for several other indications, e.g., as a dressing in burn patients, and for the reconstruction of dental defects and oro-pharyngeal fistulas [24,25]. Currently, there is a growing interest in expanding the applications of the human amnion, due to its wide availability, low cost, and interesting regenerative properties. The anti-inflammatory and neuroprotective properties of the amnion could be used in the context of radical prostatectomy. In animal experiments, Lemke et al. showed the prevention of nerve fibrosis via amnion wrapping after thermal damage to the rat’s sciatic nerve [18]. A further Chinese experimental study showed improved sciatic nerve regeneration after pressure damage and wrapping with a PLCL-AM membrane. Immunohistochemistry showed reduced macrophages, collagen formation, and an increased detection of nerve growth factor [26]. Other experimental and small clinical studies used the amnion for nerve regeneration; for example, in thyroid surgery and carpal tunnel syndrome [27,28]. In a retrospective evaluation of the use of the dehydrated amnion/chorion (AmnioFix) as part of a nerve-sparing robotic radical prostatectomy (RP), an American study was able to show a faster return to potency and continence without any increased risk of recurrence at a median follow up of 4 months [29]. The mean time to achieve continence (AM 1.21 months versus control group 1.83 months; *p* = 0.033) and the mean time to potency were improved in the amniotic group (AM 1.34 months versus control group 3.39 months; *p* = 0.007). Another group compared 1400 patients who underwent complete bilateral nerve-sparing robotic-assisted RP conducted by one surgeon, with 700 patients receiving an AM allograft around the NVB, with a matched control group of 700 patients, retrospectively, over a 1-year period [30]. Binary logistic regression showed that the AM was an independent significant (*p* < 0.001) predictor of achieving potency at 1 year. The amnion group was 3.86 times more likely (95%, CI 2.43–6.13) to become potent at 12 months than the control group. Following the concept from bench to bedside, we developed a tissue-engineering strategy to improve nerve regeneration after the surgical trauma of prostatectomy. After the preliminary in vitro and animal studies, we started to use amnion and methylene blue during radical prostatectomy. We used a dehydrated amnion as an overlay on NVB and dorsal anastomosis in radical prostatectomy, as previously described by Ogaya-Pinies [9]. In contrast to previous studies, we adapted the position of the AM to the visualization of the nerve plexus using methylene blue. We used the technique described by Ralis et al. [31]. The authors identified the concentration 2:8 (methylene blue:saline solution) as the ideal concentration, strong enough to stain the nerve fibers clearly, but weak enough to avoid staining the surrounding tissue. This method faded into obscurity, and has not been commonly applied in clinical use. A German research group used MB in the Göttinger minipig model to stain the nerve bundle intraoperatively [20]. The stained and visually identified nerve structures could be spared, preserving urinary bladder function. That the nerve structures can be easily damaged was demonstrated on the contralateral side, where no methylene blue staining was used. Additionally, they demonstrated that the dissection of the colored structures led to a total loss of organ function. The histologic investigation of these bundles revealed them to be autonomous nerve fibers [20]. Additionally, methylene blue has emerged as a potential drug in the treatment of neurodegenerative diseases, such as Alzheimer’s, Parkinson’s, and Huntington’s diseases, traumatic brain injury, etc., owing to its cognitive improvement and neuroprotective functions [32,33]. These functions are based on the anti-oxidative and anti-inflammatory effects of MB, and its ability to prevent autophagy [34,35]. We wanted to use these functions to reduce posttraumatic nerve degeneration. Our trial has several limitations, due to the small patient number and retrospective design. A major obstacle to the recognition of the AM as a promising material for surgery is the low quality of published studies, and the lack of a translational approach. In our work, however, the AM has been used as a bioactive scaffold, both experimentally in animals and clinically, to create a suitable environment for regeneration. Nerve regeneration is inherently limited, and the AM alone can only improve the outcome, not guarantee complete regeneration and functional restoration. We should be aware of the additional costs. The material costs of open radical prostatectomy in our department are EUR 164, and the cost of the amnion and methylene blue is about EUR 400. In contrast to previous amnion studies, we carefully selected our patients. We aimed to improve early continence and VUA healing, which we could prove through pad usage and cystoscopy. In addition to improving functional outcomes through neuroprotection, we aim to explore the anti-carcinogenic effect in future trials, measured using biochemical recurrence [19]. Unlike in the USA, we can only use off-label amnions in urological indications. In the opinion of the PEI, none of the amnion preparations currently approved in Germany justify use in the context of prostatectomy. A clinical trial would have to be conducted in the future, according to § 21 para. 4 AMG. Another important problem in tissue engineering is the lack of materials in clinical practice. One main reason could comprise the long transfer time and the diverse preclinical knowledge required, starting with cell culture and in vitro studies, then in vivo animal models, leading to clinical trials and commercialization, with regulatory oversight at all stages [36]. The introduction of new surgical methods, innovations, or variations does not yet follow clear, standardized paradigms. Innovation, Development, Exploration, Assessment, and Long-term Study (IDEAL) is a new reporting approach introduced by an international panel of surgeons, researchers, editors, statisticians, and other stakeholders committed to the production, dissemination, and evaluation of quality research in surgery [37]. We aim to follow the IDEAL recommendations for surgical innovation in order to obtain replicable best evidence in the future. The current study represents the development step of IDEAL. Our surgical innovation was raised from preliminary animal studies. After the first published animal and clinical studies, we added iterations, such as the methylene blue visualization, to improve outcome. This publication describes a first-in-human concept, where we standardize the procedure. 

The study’s limitations are the low patient number and the short-term outcome that might not reflect the most important effects of the procedure, and the data are insufficient to allow any meaningful statistical analysis. However, we believe that it is important to publish the results for our development stage of the IDEAL study. We report transparent sequential cases, and can standardize the surgical procedure and define the key elements for future studies [38]. For these reasons, we decided to report our early results. Several iterations of the procedure were conducted. During the initial two cases, we placed amniotic tissues before the anastomosis suturing. Due to some bleeding, and suction being needed, there was a risk of amnion displacement. We changed the procedure, and placed the amnion after the anastomosis sutures, before making the knot. Two planned cases were excluded, due to unexpected extended disease. In the next step, we will identify a subgroup of patients who will benefit most from the innovation, and undertake a prospective propensity-score-matched comparison to a standard procedure. Additionally, long-term oncological outcomes will be evaluated.

## 5. Conclusions

Strategies to prevent fibrotic reactions within the urinary tract, and improve nerve healing in reconstructive urology are lacking. A combination of the AM and MB can be used as immunomodulatory tissue engineering for neuroprotection during radical prostatectomy in selected cases. The application was shown to be feasible, with improved early continence. Following this, future research following the IDEAL system, examining the potential of the AM and MB in the context of radical prostatectomy, is warranted.

## Figures and Tables

**Figure 1 biomedicines-11-02260-f001:**
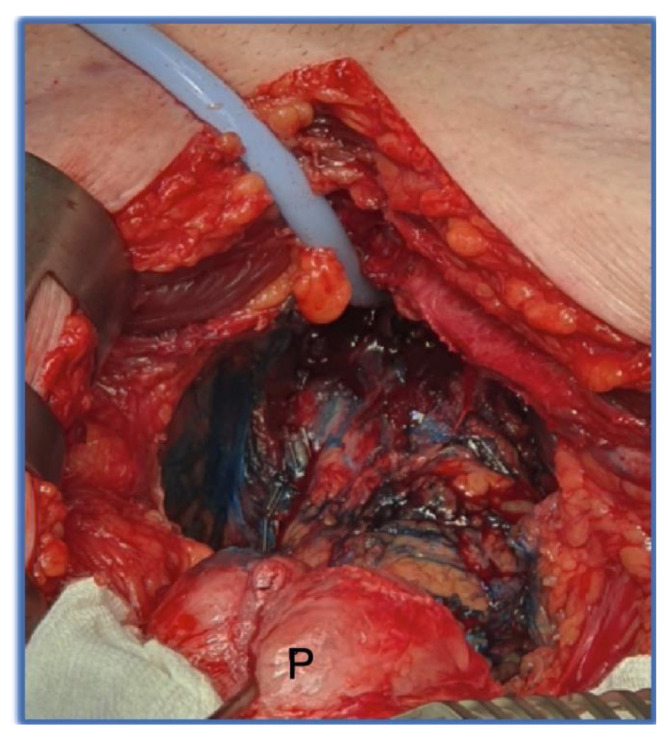
Step 1 of tissue engineering: visualization of the pelvic nerve plexus with MB. View from above into the pelvis during retropubic prostatectomy. The prostate (P) was dissected from the urethra (catheter), and tilted upward.

**Figure 2 biomedicines-11-02260-f002:**
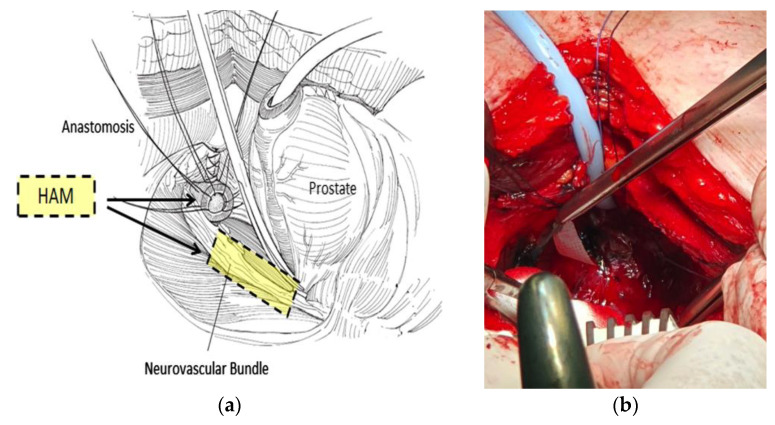
Step 2 of tissue engineering: the dehydrated HAM is cut into three pieces, and placed over the neurovascular bundles, and at the vesicourethral anastomosis. (**a**) Schematic diagram. (**b**) Intraoperative image of an open retropubic prostatectomy; the prostate has been removed, anastomotic sutures have been presented, and an amniotic patch is placed in the area of the left NVB [11].

**Figure 3 biomedicines-11-02260-f003:**
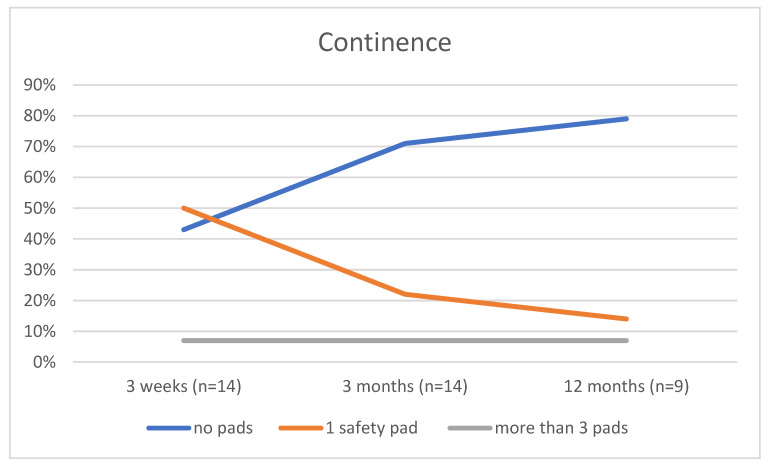
Postoperative continence over time (3 weeks, 3 months, 12 months) presented as pad usage over 24 h.

**Table 1 biomedicines-11-02260-t001:** Preoperative patient characteristics.

Patient Preoperative Characteristics	*n* = 14
Age	66 (56–74)
Smoking	
current	4 (29%)
former	2 (14%)
never	8 (57%)
Diabetes mellitus	0
ASA score	2 (1–3)
PSA	5.75 (4.5–10.8)
D’Amico classification	
low risk	2 (14%)
intermediate risk	11 (79%)
high risk	1 (7%)
Continence	14 (100%)
Erectile function sufficient for intercourse	
no sexual activity	2 (14%)
≥50%	12 (86%)

Data are presented as the median (range) for continuous variables, and *n* (%) for categorical variables. ASA = American Society of Anesthesiologists.

**Table 2 biomedicines-11-02260-t002:** Intra- and postoperative parameters.

Intra- and Postoperative Parameters	*n* = 14
Follow up, months	11 (±4.6)
Operating time, min	71 (56–85)
Transfusion	0
Nerve sparing	
unilateral	1 (7%)
bilateral	11 (79%)
no	2 (14%)
Catheterization time, days	8 (6–27)
Hospital stay, days	8 (7–14)
**Complications**	
obstructive voiding	2 (14%)
urgency	1 (7%)
recurrent urinary tract infections	1 (7%)
**Continence**	
no pads	11 (79%)
one safety pad	2 (14%)
more than five pads	1 (7%)
**Erectile function sufficient for intercourse**	
no sexual activity	5 (36%)
no intercourse possible	4 (28%)
50% (with pills, pump, or injection)	5 (36%)
**Oncological outcome**	
Gleason score ≤ 6	1 (7%)
Gleason score 7	12 (86%)
Gleason score ≥ 8	1 (7%)
extracapsular extension	2 (14%)
seminal vesicle invasion	1 (7%)
positive margins	3 (21%)
lymph node invasion	1 (7%)
adjuvant radiotherapy	2 (14%)
adjuvant hormone therapy	1 (7%)
Would you undergo the surgery again?	14 (100%)

Data are presented as the median (range) for continuous variables, and *n* (%) for categorical variables. The follow up is presented as the mean and standard deviation.

## Data Availability

The data presented in this study are available on request from the corresponding author. The data are not publicly available due to privacy reasons.

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
