# Peer review of "Functional Recovery after the Application of Amniotic Tissues and Methylene Blue during Radical Prostatectomy—A Pilot Study"

_biomedicines, 2023, doi:10.3390/biomedicines11082260_

Round 1

Reviewer 1 Report

The authors report the use of both methylene blue to identify NV bundles, and AM to potentially protect the fibers and reduce damage intra op and post  op.

14 pts are reported

Essentially this is a pilot study. The authors would do well to discuss a stastistical rationale/end point nad justification for the sample size. The study is not a phase I as toxicity is not assessed in different dose/application levels.

Other than unclear rationale for study numbers, the endpoints are well characterized.

the authors do not suggest overly broad conclusions, and acknowledge a randomized trial would be needed to prove the potential benefits of the approach.

Author Response

Thank you very much for your review and suggestions.

We corrected the title to highlight that this a pilot study and added the discussion on the sample size:

This study aims to improve the surgical technique and not to develop a pharmaceutical. Our surgical innovation raised from preliminary animal studies. After the first published animal and clinical studies we added iterations like methylene blue visualization to improve an outcome. In this publication we describe a first-in-human concept where we stabilize the procedure. Innovations undergo rapid iterative change in the light of experience. Therefore, it is the development stage that most clearly differentiates the pathway for surgery innovation from that for pharmaceutical innovations. In both a scientific and ethical sense, development is the most problematic of the stages, and as a result is often poorly reported. Judgments about success or failure at this stage may be made on the basis of short term outcome measures that might not reflect the most important effects of the procedure, and frequently the data are insufficient to allow any meaningful statistical analysis. Key elements are a prior protocol, clearly defined objective outcomes, and transparent sequential reporting of cases, showing when changes in indication or technique are made. However, data from this type of study will be more reliable and valid than information obtained from retrospective series [37]. Due to these reasons we aimed to report the early results. Several iterations of the procedure were done. During the initial two cases, we placed amniotic tissues before the anastomosis suturing. Due to some bleeding and suction needed, there was a risk of amnion displacement. We changed the procedure and placed amnion after the placement of the anastomosis sutures and before making the knot. After 5 cases we additionally placed Tabotamp fibrillar (Ethicon, USA) on the neuro-vascular bundles in two cases with extended bleeding. Two planned cases were excluded due to unexpected extended disease. In the next step we will identify a subgroup of patients which will benefit most from the innovation and undergo a propensity-score matched comparison to a standard procedure. Additionally long term oncological outcomes will be evaluated. In the next exploration step a prospective comparison study is planned.

McCulloch P, Cook JA, Altman DG, Heneghan C, Diener MK; IDEAL Group. IDEAL framework for surgical innovation 1: the idea and development stages. BMJ. 2013;346:f3012.

Reviewer 2 Report

The authors presented the interesting concept of improving functional results of RP. However several doubts were noticed in the paper: 

-       line 26 ‘the postoperative continence rate, defined as no pads used, varies between 4% and 31%, mean 16%’ while the current data brings the image of nearly 100% continence post robotic RP. 

-     line 31  ‘However, laparoscopic and robotic assisted RP with nerve-sparing strategies have failed to show significant benefits in terms 32 of oncological and functional outcomes compared to open RP [4]’ the existing data are contrary.

-      line 33 ‘Surgical intervention is 33 main reason for scarring and nerve damage [5]’ please note the devastating effect of radiotherapy.

-       Lines 49-80: the paper lacks the aim while this section covers discussion and conclusions

-       Line 83: What were the selection criteria?

-       The first major concern is the methodology: the data of the experimental group should be compared with other RP cases.

-       The second major concern: can you specify Bioethics committee number? Why did you name the study retrospective? Did you start the procedures With no approval of the committee?

-       Where the procedures performed via open approach? Why?

Author Response

The authors presented the interesting concept of improving functional results of RP. However several doubts were noticed in the paper: 

-       line 26 ‘the postoperative continence rate, defined as no pads used, varies between 4% and 31%, mean 16%’ while the current data brings the image of nearly 100% continence post robotic RP. 

 The text was changed accordingly. Even better early continence results are presented by the robotic series.

-     line 31  ‘However, laparoscopic and robotic assisted RP with nerve-sparing strategies have failed to show significant benefits in terms 32 of oncological and functional outcomes compared to open RP [4]’ the existing data are contrary.

We don´t completely agree with the reviewer. The reviewer provides no references. The available randomized studies  and meta-analyses show partly a benefit for the early continence and potence but there is a lack of evidence for a clear benefit. A current meta-analysis found no significant difference between robotic and laparoscopic approach regarding continence (odds ratio [OR] 1.95, 95% confidence interval [CI] 0.67-5.62) after 12 mos. However, at 3 mo and 6 mo there were significant differences in favour of RARP. Potent patients undergoing RARP consistently show better potency postoperatively, compared to laparoscopic approach.

Haney CM, Kowalewski KF, Westhoff N, et al. Robot-assisted Versus Conventional Laparoscopic Radical Prostatectomy: A Systematic Review and Meta-analysis of Randomised Controlled Trials [published online ahead of print, 2023 Jun 21]. Eur Urol Focus. 2023;S2405-4569(23)00118-9.

-      line 33 ‘Surgical intervention is 33 main reason for scarring and nerve damage [5]’ please note the devastating effect of radiotherapy.

 We added the statement on radiotherapy. 2 patients underwent adjuvant radiotherapy, both patients were continent.

-       Lines 49-80: the paper lacks the aim while this section covers discussion and conclusions.

Thank you. We added the aims. The aim of this proof-of-concept study was to stabilize the procedure and report on iterations and the feasibility of the technique modifications.

-       Line 83: What were the selection criteria?

Thank you, we added the selection criteria. Preoperatively continent patients (no pads used), between 40 and 75 years old, with a clinical localized PCa (cT1- T2) were included. The exclusion criteria were: preoperative erectile dysfunction (ED, not enough for sexual intercourse in < 50%), patients younger than 40 or older than 75 years, preoperative signs of extracapsular disease (cT3 or signs of extracapsular extension on imaging) or any previous prostatic surgery or prostate cancer treatment. Also excluded were patients lost to follow-up, not in compliance, or those having concurrent cancer diagnosis or important medical conditions.

-       The first major concern is the methodology: the data of the experimental group should be compared with other RP cases.

 As we stated previousely a comparison with the matched RP cases is planned in a next step, when we identify the subgroup of interest.

-       The second major concern: can you specify Bioethics committee number? Why did you name the study retrospective? Did you start the procedures With no approval of the committee?

As we stated in the discussion part of the paper this is a retrospective study, which has been proved and approved by PEI (reference: NIS 524), federal drug approval authority. An additional approval by ethics committee is not mandatory.

-       Where the procedures performed via open approach? Why?

Open approach is a standard procedure in our institution.

Reviewer 3 Report

An interesting study combining 2 different approaches for neuroprotection during RP - MB (as a agent helping in visualization of NVB and as an neurotropic agent) and amniotic membrane 

Title - a very good example for proof-of-concept study - in this reviewer`s opinion this should be emphasized in the title - Major

Introduction - concise. nicely written paragraph.

Material and methods - sophisticated description study design and surgical procedure

Results - nicely presented and visualized 

Discussion - several points need additional discussion according to this reviewer`s opinion:

- lack of control group - although there is enormous body of data to be used, the optimal method will be the results from the studied procedure to be compared with a control group from the same center - Major

- short follow-up in significant percentage of the patients - median follow-up time could be misleading in a limited sample size - mean value is preferable - Major

Regarding all the aforementioned, my recommendation is to reconsider this manuscript for publication after satisfactory comments and appropriate modification by the authors on the aforementioned issues

Author Response

Thank you very much for the review and the valuable comments. The title was changed. We added the discussion on the sample number and the control group. We changed the follow up to mean values.

This study aims to improve the surgical technique and not to develop a pharmaceutical. Our surgical innovation raised from preliminary animal studies. After the first published animal and clinical studies we added iterations like methylene blue visualization to improve an outcome. In this publication we describe a first-in-human concept where we stabilize the procedure. Innovations undergo rapid iterative change in the light of experience. Therefore, it is the development stage that most clearly differentiates the pathway for surgery innovation from that for pharmaceutical innovations. In both a scientific and ethical sense, development is the most problematic of the stages, and as a result is often poorly reported. Judgments about success or failure at this stage may be made on the basis of short term outcome measures that might not reflect the most important effects of the procedure, and frequently the data are insufficient to allow any meaningful statistical analysis. Key elements are a prior protocol, clearly defined objective outcomes, and transparent sequential reporting of cases, showing when changes in indication or technique are made. However, data from this type of study will be more reliable and valid than information obtained from retrospective series [37]. Due to these reasons we aimed to report the early results. Several iterations of the procedure were done. During the initial two cases, we placed amniotic tissues before the anastomosis suturing. Due to some bleeding and suction needed, there was a risk of amnion displacement. We changed the procedure and placed amnion after the placement of the anastomosis sutures and before making the knot. After 5 cases we additionally placed Tabotamp fibrillar (Ethicon, USA) on the neuro-vascular bundles in two cases with extended bleeding. Two planned cases were excluded due to unexpected extended disease. In the next step we will identify a subgroup of patients which will benefit most from the innovation and undergo a propensity-score matched comparison to a standard procedure. Additionally long term oncological outcomes will be evaluated. In the next exploration step a prospective comparison study is planned.

McCulloch P, Cook JA, Altman DG, Heneghan C, Diener MK; IDEAL Group. IDEAL framework for surgical innovation 1: the idea and development stages. BMJ. 2013;346:f3012.

Round 2

Reviewer 2 Report

Although the authors responded to some of my comments, I still have doubts concerning the publication of the paper. 

Author Response

Thank you very much, additional revision and editing have been done.

Reviewer 3 Report

the authors have sufficiently address the issue raised from the previous review.

it is this reviewer opinion that the manuscript should be published in the present form

Author Response

Thank you very much.